# A Dual Role of Heme Oxygenase-1 in Cancer Cells

**DOI:** 10.3390/ijms20010039

**Published:** 2018-12-21

**Authors:** Shih-Kai Chiang, Shuen-Ei Chen, Ling-Chu Chang

**Affiliations:** 1Department of Animal Science, National Chung Hsing University, Taichung 40227, Taiwan; shihkaichiang@gmail.com (S.K.C.); chenshuenei@hotmail.com (S.E.C.); 2Innovation and Development Center of Sustainable Agriculture (IDCSA), National Chung Hsing University, Taichung 40277, Taiwan; 3The iEGG and Animal Biotechnology Center, National Chung Hsing University, Taichung 40277, Taiwan; 4Research Center for Sustainable Energy and Nanotechnology, National Chung Hsing University, Taichung 40277, Taiwan; 5Chinese Medicinal Research and Development Center, China Medical University Hospital, Taichung 40447, Taiwan

**Keywords:** ferroptosis, heme oxygenase-1, iron, reactive oxygen species, glutathione, chemotherapy

## Abstract

Heme oxygenase (HO)-1 is known to metabolize heme into biliverdin/bilirubin, carbon monoxide, and ferrous iron, and it has been suggested to demonstrate cytoprotective effects against various stress-related conditions. HO-1 is commonly regarded as a survival molecule, exerting an important role in cancer progression and its inhibition is considered beneficial in a number of cancers. However, increasing studies have shown a dark side of HO-1, in which HO-1 acts as a critical mediator in ferroptosis induction and plays a causative factor for the progression of several diseases. Ferroptosis is a newly identified iron- and lipid peroxidation-dependent cell death. The critical role of HO-1 in heme metabolism makes it an important candidate to mediate protective or detrimental effects via ferroptosis induction. This review summarizes the current understanding on the regulatory mechanisms of HO-1 in ferroptosis. The amount of cellular iron and reactive oxygen species (ROS) is the determinative momentum for the role of HO-1, in which excessive cellular iron and ROS tend to enforce HO-1 from a protective role to a perpetrator. Despite the dark side that is related to cell death, there is a prospective application of HO-1 to mediate ferroptosis for cancer therapy as a chemotherapeutic strategy against tumors.

## 1. Introduction

Oxidative stress is caused by an imbalance between cellular oxidants and antioxidants. Reactive oxygen species (ROS) are the major cellular oxidants, which are normally generated as by-products in oxygen metabolism. However, under some circumstances, extracellular insults (e.g., ionizing radiation and UV light), xenobiotics, and pathogens also greatly provoke ROS production, leading to an imbalance of the intracellular reduction-oxidation (redox) status [1]. Excessive ROS can induce oxidative damage of DNA, and, to a higher degree, gene mutation and carcinogenesis [2,3,4]. Moreover, lipid peroxidation by excessive ROS may damage cellular structures and eventually induce cell death [1]. In fact, the augmentation of ROS is a useful approach for clinical cancer treatment. Various chemotherapeutic agents, such as cisplatin, doxorubicin, and 5-fluorouracil, have been shown to exert their antitumor activity via ROS-dependent activation of apoptosis [4,5]. Therefore, oxidation therapy becomes a possible strategy by provoking ROS production and diminishing antioxidant enzymes in cancer cells. Ferroptosis is a newly identified non-programmed cell death, characterized by excessive accumulation of free cellular iron and severe lipid peroxidation [6]. This ROS- and iron-overload cell death became a new therapeutic strategy in several diseases, especially in cancer treatment. Indeed, ferroptosis-inducing agents (erastin, RSL3, and sorafenib) have demonstrated therapeutic effects against cancers [6,7].

Heme oxygenase-1 (HO-1) is a phase II enzyme that responds to electrophilic stimuli, such as oxidative stress, cellular injury, and diseases. HO-1 is elevated in various human malignancies, implicating its contribution to settle the tumor microenvironment for cancer cell growth, angiogenesis, and metastasis, as well as resistance to chemotherapy and radiation therapy. By contrast, augmented expression of HO-1 in tumor cells can enhance cell death in many cancers. [8,9,10,11]. Its multiple pleiotropies in tumorigenesis, including tumor initiation, angiogenesis, and metastasis, have been well reviewed [8,9,10,11]. Although the bright and dark sides are both discussed in different studies, HO-1 has been widely recognized to play a cytoprotective role in tumor cells to conquer the assault of augmented oxidative stress by chemotherapeutic agents, thus preventing the cancer cells from apoptosis and autophagy, and even promoting cell proliferation and metastasis [8,9,11]. The protective or detrimental effects of HO-1 were also reported in different diseases, including kidney injury and neurodegeneration [12,13,14]. Emerging evidence has revealed another dark side of HO-1, showing that HO-1 induces ferroptosis through iron accumulation [15,16,17] or other unknown mechanisms. Based on the current findings, this review provides a brief background on the biological functions of HO-1, as well as its metabolites, namely biliverdin/bilirubin, carbon monoxide, and ferrous iron, to delineate how HO-1 mediates ferroptosis.

## 2. Ferroptosis and Cancer

Ferroptosis is a recently identified type of cell death that is morphologically, genetically, and mechanistically distinct from regulated cell death, including apoptosis, necroptosis, and autophagy [6]. Ferroptotic cells are morphologically characterized by small mitochondria, collapsed mitochondrial cristae, and increased mitochondrial membrane density [6]. Mechanistically, ferroptosis is induced by iron accumulation and lipid peroxidation, accompanied by glutathione depletion. Excessive lipid peroxidation impairs the cellular membrane fluidity, permeability, and cellular integrity, eventually leading to cell death [6,18,19,20]. Ferroptosis can be induced by the overloading of iron (ferric ammonium citrate), glutathione/glutamine antiporter system Xc^−^ inhibition (e.g., erastin, sorafenib, sulfasalazine, and lanperisone), and glutathione peroxidase 4 (GPx4) inactivation (RSL3, DPI7). Pharmacological manipulations, such as iron chelators (deferoxamine, ciclipirox, and deferiprone), glutathione replenishment (*N*-acetyl-l-cysteine, β-mercaptoethanol, cysteine/cystine, intracellular glutathione), and inhibitors of ROS production and lipid peroxidation (liporxstatin-1, ferrostatin-1, zileuton) that modulate the ferroptotic process have been shown to function against diseases, including cancer, neurotoxicity, and ischemia/reperfusion-induced injury [18,19,20]. 

The interrelationship between ferroptosis and cancer progression has been validated using ferroptotic agents. Some small molecules (e.g., erastin and RSL3) and clinical cancer drugs (e.g., sorafenib, sulfasalazine, and artesunate) induced cell death via the inhibition of system Xc^−^ and GPx4 in various types of cancer cells [6,7,19]. A delay of ferroptosis protects cancer cells from metabolic oxidative stress, and thus increases their survival and distal metastasis [21]. Besides, the induction of ferroptosis was shown to overcome artesunate-induced resistance in head and neck cancer cells [22], and the induction of ferroptosis that contributes to anticancer activity has been identified in different cancer types [23]. Diffuse large B-cell lymphomas and renal cell carcinomas strongly rely on GPx4 availability to maintain redox status, and thus may suggest a high sensitivity to ferroptosis [7]. Certain cancer cells, such as pancreatic cancer cell lines (MIA PaCa-2, PANC-1, and BxPC-3) and a subset of triple-negative cancer cells also greatly depend on system Xc^−^ to mediate cysteine uptake for growth, as well as survival under oxidative stress conditions [24,25], suggesting that system Xc^−^ might serve as a good chemotherapeutic target. These results delineate the potential of ferroptotic process in clinical applications. 

## 3. HO-1-mediated Ferroptosis in Cancer Cell Survival

As a dual regulator in iron and ROS homeostasis [8,26,27], HO-1 was suggested to serve a dominant role in ferroptosis [15,16,17,22,28,29,30,31]. Alzheimer’s patients exhibited enhanced lipid peroxidation, which may be associated with HO-1 elevation and iron accumulation [32]. In HT-1080 fibrosarcoma cells, erastin induces a time- and dose-dependent increase of HO-1 expression [15]. Evidences from HO-1 knockdown mice and by the use of HO-1 inhibitor zinc protoporphyrin IX showed that HO-1 promotes erastin-induced ferroptosis and it is associated with iron bioavailability, but not with biliverdin and bilirubin [15]. However, HO-1 also functions as a negative regulator in erastin- and sorafenib-induced hepatocellular carcinoma since knockdown of HO-1 expression enhanced cell growth inhibition by erastin and sorafenib [31]. A similar result of HO-1 to ameliorate ferroptosis induction was also observed in renal proximal tubule cells [28]. Immortalized renal proximal tubule cells that were obtained from HO-1^−/−^ mice exhibited more pronounced cell death induced by erastin and RSL3 than those from wild type mice [28]. In contrast to the negative role in ferroptosis, several recent studies have demonstrated that enhanced HO-1 expression can augment or mediate anti-cancer-agent (Bay117085 and withaferin A) induced ferroptosis by promoting iron accumulation and ROS production [16,17]. Genetic knockdown and the pharmacological inhibition of HO-1 also validated that HO-1 activation triggers ferroptosis through iron overloading and subsequently excessive ROS generation and lipid peroxidation [16,17]. The silencing of HO-1 by siRNA also reversed the resistance to artesunate-induced ferroptosis in cisplatin-resistant head and neck cancer cells [22]. Based on the contradictory results, it appears that HO-1 activation as a cytoprotective defense or governing ferroptotic progression depends on the degree of ROS production and following oxidative damage in response to stimulatory cues.

## 4. HO-1 Activation and Heme Metabolites

Heme oxygenases, including HO-1 [also called heat shock protein 32 (Hsp32)], are rate-limiting enzymes in the breakdown of heme (iron protoporphyrin IX). Degradation of heme produces biliverdin, carbon monoxide (CO), and iron (ferrous iron, Fe^2+^) (Figure 1). Biliverdin is subsequently converted to bilirubin by biliverdin reductase. Oxygen, nicotinamide adenine dinucleotide phosphate (NADPH), and cytochrome p450 reductase are required in this catalytic reaction [26,27]. Cellular iron accumulation upregulates the expression of ferritin, which sequesters the pro-oxidant effect of iron [33]. Three types of heme oxygenases are found in mammalian cells, the inducible form HO-1, constitutive form HO-2, and HO-3, which is mostly inactive. HO-1 can be induced by a wide spectrum of cues, including oxidants, inflammatory mediators, chemicals, physical stimuli, and its own substrate, heme [26,27]. After synthesis, the HO-1 protein is normally anchored in the endoplasmic reticulum [34]. The subcellular location of HO-1 is dynamic. Some pathogenic stimuli may induce the translocation of HO-1 into the plasma membranes, nucleus, and/or mitochondria, which might allow the enhancement of HO-1 activity for heme degradation within the target compartment [35,36].

The most important activation of HO-1 is mediated by nuclear factor erythroid 2-related factor 2 (Nrf2). Under resting conditions, Nrf2 activity is inhibited by physical interaction with Kelch-like ECH-associated protein 1 (Keap1), leading to the recruitment of Cullin-3-dependent E3 ubiquitin ligase for proteasomal degradation, thereby maintaining Nrf2 at a low level [37]. Under oxidative stress, Keap1 undergoes a conformational change and releases Nrf2. Free Nrf2 then translocates into the nuclei where it interacts with small Maf protein and further binds onto the antioxidant-response element (ARE) or electrophile-response element (EpRE), to transactivate various genes encoding antioxidant enzymes, including HO-1 [38]. Increased cellular heme level hampers the induction of HO-1 through Bach1, a Nrf2 antagonist, due to the competition for the promoter binding site [39]. Depletion of cellular glutathione has been shown to increase HO-1 gene transcription in the mouse motor neuron-like hybrid cells, NSC34 cells [40]. HO-1 abundance is also regulated by an endoplasmic reticulum-associated degradation pathway [41]. In HIV-infected astrocytes, HO-1 was degraded in an immunoproteasome-dependent pathway in response to IFNγ and TNFα/LPS stimulation [42].

The pleiotropic effects of HO-1 and metabolites from heme on tumor growth, neurodegenerative diseases, ischemia/reperfusion injury, and renal injury have been thoroughly reviewed [8,9,11,13,14]. Most of the evidence has suggested that HO-1 functions in cytoprotective defense mechanisms against oxidative attacks through its metabolites biliverdin/bilirubin and CO. However, those metabolites also have demonstrated the detrimental effects, especially in neuronal damage and degeneration [14]. Both biliverdin and bilirubin can inhibit the peroxidation of lipid and protein through scavenging ROS [43]. Biliverdin also shows an ability to modulate the activation of endothelial nitric oxide synthase, leading to a decrease in nitric oxide production [44]. Another protective effect of biliverdin and bilirubin is to interfere with the apoptotic process [45]. Moreover, biliverdin provides a neutralizing activity of ROS, contributing to a proapoptotic effect and the suppression of tumor growth in head and neck squamous cell carcinoma [46]. The cytoprotective or detrimental effects of heme metabolites are determined by or are attributed to their intracellular levels. A high concentration of biliverdin has been shown to cause apoptosis in cancer cells [47]. Overproduction of bilirubin by hemolytic hyperbilirubinemia is associated with bilirubin neurotoxicity in newborns [48].

Another heme metabolite, CO, a gaseous product, is an important signaling molecule, possessing the vasodilatory, anti-inflammatory, anti-proliferative, anti-apoptotic, thrombosis, and angiogenesis activities in various cell types [8,9]. The mechanisms of intracellular events impacted by CO are complicated. CO also exerts both beneficial and deleterious effects, depending on its targeted molecules. CO can activate soluble guanylyl cyclase, followed by cGMP generation, linking cellular proliferation, thrombosis, and vasodilation. CO can also modulate single kinases, including p38 MAP kinase, ERK, and JNK. The activation of p38 can lead to the downregulation of pro-inflammatory cytokines and the upregulation of anti-inflammatory cytokine production, contributing to the anti-inflammatory protection of tissue [8,9]. CO can cooperate with NF-κB to modulate the expression levels of several anti-apoptotic proteins [8,9].

The last metabolite of HO-1, ferrous iron, is toxic due to the ability to interact with cell oxidants to generate ROS [49]. The details are discussed in the next section. In addition to the significant impact on signaling pathways by heme metabolites, HO-1 can mediate various signaling pathways per se, rather than depending on the enzyme activity. A mutated form of HO-1 protein that is defective in catalytic activity could protect cells against oxidative injury [50]. The benefits of enhanced antioxidant activity by HO-1 were associated with increased catalase activity and glutathione levels [50]. 

## 5. HO-1 and Iron

Iron (Fe), an essential metal for biological activities, participates in electron transport of the respiration chain, heme synthesis, erythropoiesis, and enzyme systems. However, iron is a potential toxicant to cells due to its pro-oxidant activity, which can lead to oxidant DNA damage, causing neurodegenerative diseases and promoting oncogenesis [20,49,51]. Ferroptosis, a form of iron-mediated oxidative cell death, has been shown to play a critical role in the pathogenesis involving iron-overload, such as cancer and neurodegenerative diseases [18,19,20,51], thus implying a harmful role of HO-1.

Iron is involved in the transfer of electrons via oxidation-reduction reactions to transition between the ferric (Fe^3+^) and ferrous (Fe^2+^) states [50]. The same mechanism is employed during intracellular transport and toxicity production of iron. Transferrin is responsible for iron transport in the bloodstream. Iron binds to transferrin in an oxidized ferric state (Fe^3+^). Iron can enter cells by two modes, transferrin receptor-mediated endocytosis and independent transport of non-protein-bound iron (NPBI). In the NPBI system, ferrous irons diffuse into cells through binding with the low-molecular-weight complexes, such as adenosine triphosphate, citrate, ascorbate, peptides, or phosphatases [52,53]. After acidification within the endosome, iron disassociates from transferrin and is reduced by ferric reductase into ferrous iron, which is then transported into the cytosol by divalent metal transporter 1 (DMT1). In the cytosol, ferrous iron can be used, stored into ferritin, or effluxed from cells by the iron exporter ferroportin [53]. Intracellular iron mainly binds to specific proteins, such as ferritin, hemoproteins, and various iron-containing proteins for further utilization. During iron deficiency, iron-binding ferritin can undergo recycling by autophagic turnover in the lysosome [54]. Nuclear Co-Activator 4 (NCOA4) is an autophagic cargo receptor, which can bind to ferritin and carry it into the autophagosome [55].

Few irons in the cytosol are deposited in the labile iron pool where the redox-active iron (Fe^2+^) is oxidized by hydrogen peroxide to Fe^3+^ (ferric) and therefore generates ROS, including soluble radical (HO⋅), lipid alkoxy (RO⋅), and hydroxide ion (OH^−^) via the Fenton reaction. Free iron in the redox-active form is easily accessed as a pro-oxidant [49]. Thus, the NPBI system is crucial for iron overload to induce the lipid peroxidation [52,53]. Overloading of iron can promote the Fenton reaction and ROS generation [49]. Excessive ROS production consequently results in the peroxidation of adjacent lipid and oxidative damage of DNA and proteins, eventually inducing ferroptosis [6,7,18,19,20].

Cellular iron homeostasis and distribution are regulated by specific iron-regulating proteins. At a low iron level, these regulatory proteins can bind to the iron-response element of target genes to inhibit the expression of iron-binding proteins, such as ferritin and ferroportin, but increase the expression of transferrin receptor [56]. Enhanced HO-1 activity was shown to increase the cellular iron level and also promote ferritin production to sequester iron and following pro-oxidant effects in the meanwhile [33,57]. However, the iron-binding ability of ferritin is disturbed by oxidative stress, causing an uncontrolled release of iron, finally resulting in excessive accumulation of iron in the cytosol and enhancement of lipid peroxidation. The importance of iron in ferroptosis was thoroughly confirmed by increased iron uptake and the presence of iron chelators, such as deferoxamine and ciclopirox, and deferiprone [6,18,19,20]. Similarly, supplementation with an exogenous source of iron, such as ferric ammonium citrate, ferric citrate, and iron chloride hexahydrate enhanced ferroptosis [6]. Activation of iron metabolism-related proteins also contributes to ferroptosis. Upregulation of transferrin receptor 1 resulted in a higher sensitivity to ferroptosis induction [58,59]. The iron-response element binding protein 2 (IREB2) is essential for erastin-induced ferroptosis [6]. Moreover, a decrease in ferritin for iron storage may cause a free iron overload [58], thereby enhancing ferroptosis induction by stimulatory agents [6]. Increased degradation of NCOA4, a receptor cargo protein of ferritin, resulted in ferritinophagy and ferritin degradation, leading to an increase in free iron and ROS generation [60].

Ferroptosis induction by HO-1 is mediated by iron augmentation and lipid peroxidation [15,16,17]. The induction is tightly associated with oxidative stress, due to its predominant sensitivity to oxidative inducers, such as redox-active iron, heme, hemoglobin, and heme-containing proteins [61,62]. The upregulation of HO-1 can enhance heme degradation and ferritin synthesis and change the intracellular iron distribution [57].

## 6. HO-1 Modulation in Ferroptosis

Due to highly reactive iron and oxidative stress, the role of HO-1 in ferroptosis was recently re-proposed, in which both the protective and detrimental roles have been demonstrated [15,16,17,22,28,29,30,31]. Thus, HO-1 may provide redox-active iron production for the re-modification of biomolecules and structures, including lipid and protein peroxidation. The pro-oxidative activity of HO-1 contributes to ferroptosis induction [15,16,17,29], which relies on iron accumulation, not biliverdin/bilirubin [15]. Under pro-oxidant circumstances, more iron is released from iron-storing proteins, further increasing the production of ROS to accelerate oxidative stress. It can be explained that a moderate level of HO-1 activation exerts a cytoprotective effect, while the over-activation of HO-1 becomes cytotoxic due to excessive increase of labile Fe^2+^ behind the buffering capacity of ferritin [4,17,63]. A cascade of increased iron release and ROS production resulting in extensive oxidative damage to cells has been reported [64], exemplifying the potential role of HO-1 in iron accumulation and ROS generation, followed by lipid peroxidation and ferroptosis induction. Intriguingly, HO-1 is also transcriptionally upregulated by lipid peroxidation products (4-hydroxynonenal) and phospholipase metabolites (diacylglycerol and arachidonic acid), for example, in response to UV light induction [65]. 

Several small molecules have been identified to trigger ferroptosis through the regulation of HO-1 expression and activity (Table 1, Figure 2). These small molecules possess similar properties. Most of them can trigger iron release and generate massive production of ROS. Some molecules have been identified to induce HO-1-associated ferroptosis, including heme [66], erastin/sorafenib/RSL3 [15,22,31], magnesium isoglycyrrhizinate [29], BAY117085 [16], and withaferin A [17].

### 6.1. Heme

Heme is a large complex, comprising iron and protoporphyrin IX. Heme is regarded as a pro-oxidant molecule that participates in the formation of oxidative radicals and leads to oxidative injury [67]. However, because of free iron with different oxidation states, heme-containing enzymes can exert its catalysis in both reductive and oxidative reaction [67]. Therefore, several heme-containing proteins are involved in the electron transport chain in mitochondria and redox reactions, such as catalase, peroxidase, and cytochrome p450. Heme can also transfer and store oxygen while it binds to globin [67]. Hemolysis causes heme release into the circulation, leading to the generation of ROS and oxidative stress [67]. In plasma, free heme can be scavenged by hemopexin or degraded by HO-1 to generate biliverdin, CO, and ferrous iron [68]. Nevertheless, in severe hemolysis, the extensive release of heme from hemoglobin or decreased hemopexin level leads to an increase in the iron level in the circulation. Heme induces apoptosis and necroptosis in red blood cells and cerebral microvascular cells by increasing the cellular calcium level and depleting the glutathione reservoir [69]. These observations suggest that heme serves as a pro-oxidant molecule, not only functioning as an activator of HO-1, but also acting on the ferroptotic process.

Hemin, an iron-containing porphyrin with chlorine, promotes erastin-induced ferroptotic cell death in a HO-1-dependent manner in HT-1080 fibrosarcoma cells [15]. Similar to hemin-induced ferroptosis, iron-induced ferroptosis through increased production of ROS is also observed in neuroblastoma IMR-32 cells [17]. Interestingly, HO-1 induction might protect cells from oxidative assaults. In cultured alveolar epithelial cells, HO-1 overexpression increased the expression of ferritin and transferrin receptor, resulting in the alteration of the intracellular iron distribution and providing a protective effect against iron cytotoxicity from heme degradation [70]. Additionally, the protective effect by heme was demonstrated in oxidant hydrogen peroxide-treated human lung adenocarcinoma A549 cells. HO-1 upregulation by hydrogen peroxide enhanced heme synthesis and the expression of ferritin and transferrin receptor, leading to the capture of intracellular redox-active iron and the elimination of oxidative stress [59].

### 6.2. Erastin, Sorafenib and RSL3

Erasin, sorafenib, and RSL3 are known as ferroptosis-inducing agents to inhibit system Xc^−^ and GPx4 [18,19,20]. Both erastin and RSL3 were shown to upregulate HO-1 in renal proximal tubule cells [28] and HT-1080 fibrosarcoma cells [15]. In renal proximal tubule cells, HO-1^−/−^ cells showed more sensitivity to erastin- and RLS3-induced cell death than HO-1^+/+^ cells [28]. Thus, HO-1 played an anti-ferroptotic role due to its anti-oxidant effect against ROS. By contrast, in HT-1080 fibrosarcoma cells, HO-1^+/+^ cells exhibited a more profound effect on ferroptotic induction than HO-1^−/−^ cells. The metabolites of heme by HO-1, including iron and CO, contribute to the pro-ferroptotic effect of HO-1 [15]. In hepatocellular carcinoma, HepG2 and Hepa1-6 cells, genetic knockdown of Nrf2, HO-1, quinone oxidoreductase 1 (NQO-1), or ferritin also promoted erastin- and sorafenib-induced ferroptosis [31]. In hepatocellular carcinoma cells, Nrf2 expression contributed to ferroptosis resistance, in association with its downstream effector proteins, such as HO-1, NQO-1, and ferritin. HO-1 exerts a detoxification and antioxidant effect in mediating the anti-ferroptosis of Nrf2 [31]. Additionally, Nrf2, HO-1, and ferritin are transcriptionally regulated by p62, a ubiquitin binding protein that is involved in cell signaling pathways, such as the oxidative response and autophagy [71,72]. It is particularly noteworthy that the p62-Keap1-Nrf2 pathway plays an important role in ferroptosis through the upregulation of multiple genes, including HO-1, NQO-1, and ferritin [31]. The protective role of Nrf2-derived genes in erastin-induced ferroptosis was also proposed in glioblastoma [30]. 

### 6.3. Magnesium Isoglycyrrhizinate

Magnesium isoglycyrrhizinate (MgIG) is a hepatoprotective drug with a potential to alleviate the inflammation and accelerate the recovery of injured liver [73]. Sui et al. found that MgIG markedly attenuated liver injury and reduced fibrotic scar formation in a rat model of CCL4-induced liver fibrosis [29]. MgIG significantly inhibited the growth of hepatic stellate cells, the main effector cell in liver fibrosis [74], by promoting ferroptotic induction, as evidenced by the elevated iron accumulation and lipid peroxidation and the suppression of the cellular glutathione levels [66]. HO-1 deficiency of hepatic stellate cells HSC-T6 made them less sensitive to MgIG-induced ferroptosis. Interestingly, transferrin, transferrin receptor, and ferritin were all upregulated by MgIG in HSC-T6 cells [66]. However, MgIG ameliorated liver injury by ethanol through the inhibition of ROS and neutrophil infiltration, as well as by reduced expression of proinflammatory cytokines and chemokines [75].

### 6.4. Artesunate

Artesunate is a semisynthetic derivative of artemisinin, isolated from *Artemisia annua* L., which has been used in the treatment of *P. falciparum* malaria [76]. The main mechanism of the antimalarial action of artesunate involves NADPH activation, ROS generation, and DNA damage [77]. In MCF7 breast cancer cells, artesunate impacts the endolysosomal and autophagosomal compartments, leading to the blockade of autophagosome turnover and perinuclear clustering of autophagosomes, endosomes, and lysosomes. Free iron is thereby accumulated and serves as the major cause of ROS production, which turns out to be a critical prerequisite for artesunate-mediate cell death in MCF-7 breast cancer cells [76]. Artesunate can induce ferroptosis in head and neck cancer cells, but cisplatin-resistant cells are less sensitive to artesunate. In cisplatin-resistant head and neck cancer cells, the activation of Nrf2 and downstream targets HO-1 and NQO-1 by artesunate contributed to the resistance against ferroptosis. Inactivation of the Nrf2 pathway using a siRNA genetic approach reversed the ferroptotic induction by artesunate, and this ferroptosis resistance was further blocked by deferoxamine, an iron chelator, and by antioxidant Trolox [22]. 

### 6.5. BAY117085

BAY117085 was identified as an NF-κB inhibitor by blocking the phosphorylation and nuclear translocation of IκBα [78]. However, BAY117085 can induce ferroptosis in an NF-κB-independent manner [16]. In triple-negative breast cancer cells, MDA-MB-231 cells, and glioblastoma multiforme DBTRG-05MG cells, BAY117085 upregulated HO-1 expression through the Nrf2−SLC7A11 pathway, which, in turn, depleted the cellular glutathione reservoir and provoked ROS generation, resulting in iron accumulation and ultimately ferroptosis. Enforced expression of HO-1 substantially promoted ROS production and iron release, leading to endoplasmic reticulum stress, as evidenced by increased Chop and spliced XBP1 transcripts. Interestingly, BAY117085 also caused the compartmentalization of HO-1 within the nucleus and mitochondria, and subsequently caused mitochondrial dysfunction, leading to lysosome targeting for mitophagy [16]. Mitochondria-targeted HO-1 was further shown to induce higher ROS production, leading to mitochondrial dysfunction, such as fission and later development into cytotoxicity, as observed in macrophages, kidney fibroblasts, and chronic alcohol hepatotoxicity [79]. Mitochondrial targeting of HO-1 also enhanced autophagy by increasing the translocalization of LC3 and Drp1 into the mitochondria [79]. In doxorubicin-induced cardiomyopathy, however, mitochondrial targeting of HO-1 demonstrated a cytoprotective role to improve mitochondrial quality [80]. In contrast to that in wild-type mice, enforced expression of HO-1 remarkably ameliorated doxorubicin-mediated dilation of cardiac sarcoplasmic reticulum, mitochondrial fragmentation, and the number of damaged mitochondria in autophagic vacuoles. The amelioration was attributed to the increase in mitochondrial biogenesis, as evidenced by the upregulation of Nrf1, PGC1α, and TFAM, as well as by the attenuated changes in the expression of the mitochondrial fission mediator Fis1 and fusion mediators, Mfn1 and Mfn2 [80]. 

### 6.6. Withaferin A

Withaferin A is a steroidal lactone extracted from the roots and leaves of *Withania somnifera* Dunal, commonly known as Ashwagandha, Indian ginseng, or Indian winter cherry [81]. Due to its anti-proliferative and pro-apoptotic activities, withaferin A demonstrates its therapeutic potential for chemoprevention in various cancer types. Mechanistically, withaferin A can disturb the cell cycle, inhibit the activation of proliferation-related kinases (EGFR, Akt, and NF-κB), alter the ratio of pro-apoptotic/anti-apoptotic proteins, and provoke ROS generation. With the predominant oxidative cytotoxic effect, withaferin A was shown to cause mitochondrial dysfunction, apoptosis, and paraptosis [82]. Hassannia et al. demonstrated that withaferin A induced ferroptosis in neuroblastoma IMR-32 cells via two different pathways—repressing the protein level and activity of GPx4 and upregulating HO-1 expression [17]. Withaferin A decreases Keap1, leading to the activation of Nrf2 and upregulation of HO-1, followed by an increase in intracellular labile ferrous iron and consequently ferroptosis. Administration of withaferin A significantly suppressed the tumor growth of neuroblastoma through increased HO-1 expression and decreased GPx4 expression [17]. A detailed mechanism regarding upregulated HO-1 via directly targeting Keap1 by withaferin A has also been identified in endothelial cells [83]. 

## 7. Manipulation of HO-1 in Ferroptosis

Due to a high proliferation and fast metabolic rate, cancer cells tend to exhibit higher ROS production [2,3,4], and thus it may promote tumorigenesis and render cancer cells more vulnerable to oxidative stress-induced cell death. Manipulation of the intracellular ROS level may be a useful approach for cancer treatment [3,4]. Various ROS-modulating agents are currently developed for more precise and specific effectiveness to kill cancer cells [2]. On the other hand, cancer cells also develop redox adaption through the upregulation of anti-apoptotic and antioxidant molecules, allowing for cancer cells to survive and increase resistance against anticancer drugs. However, cancer cells remain to maintain higher ROS production than normal cells [2], and thus a dramatic increase of intracellular ROS still functions to kill cancer cells by shutting down antioxidant systems. Normal cells with a lower basal ROS level are less dependent on antioxidants, making normal cells less sensitive to oxidative insults. It is well-accepted that a moderate availability of ROS serves an oncogenic function to stimulate mutagenesis facilitating cancer cells to respond to microenvironment changes. In the meantime, the use of antioxidants should prevent tumorigenesis. By contrast, a high ROS level is toxic to cancer cells, which may induce oxidative stress to suppress cancer cell survival [2,4]. However, how to define a toxicity threshold of ROS availability to kill cancer cells or a level to help cancer cells to acquire mutations and resistance is difficult, particularly, among the various types of cancers.

Ferroptosis employs iron-dependent ROS production to kill cells [6] and HO-1 functions in ferroptosis by operating at cellular iron level and ROS generation [15,16,17]. Accordingly, when HO-1 is activated moderately, Nrf2-derived HO-1 exerts a cytoprotective effect by neutralizing ROS. Since cancer cells express a higher level of HO-1 [84,85,86,87,88,89,90], a high degree of HO-1 activation can increase labile Fe^2+^, leading to ROS overload, and thereby oxidative-cell death (Figure 3) [2,63]. The induction of ferroptosis via HO-1 activation could create a new chemotherapeutic strategy for cancer treatment. Undoubtedly, how the precise activation of HO-1 to selectively kill cancer cells and overcome drugs’ resistances needs to be defined.

## 8. HO-1 Modulators in Cancer Treatment

Despite the dual role in ferroptosis modulation, HO-1 is commonly regarded as a survival molecule, exerting an important role in cancer progression that is involved in the cell proliferation, metastasis, angiogenesis, as well as resistance against chemotherapy, radiation, and photodynamic therapy [9,11]. High levels of HO-1 expression have been shown in different cancer types [84,85,86,87,88,89,90], and the resistance to anticancer therapy was associated with a responsive upregulation of HO-1 [9,11]. Despite the facts, genetic and pharmacological approaches demonstrated the critical role of HO-1 activation in ferroptotic process in cancer cell death [16,17]. In clinical cases, patients with higher HO-1 expression showed the lower survival rate and poor outcomes [91]. Thereby, the modulation of HO-1 and HO-1 related pathway may serve as a potential therapeutic strategy. Several HO-1 inhibitors are developed and validated further. HO-1 inhibitors are categorized into two types, metalloporphyrins and imidazole-based compounds [92,93,94]. Metalloporphyrins are heme analogues composing of a protoporphyrin IX and a metal, such as zinc protoporphyrin (ZnPPIX) and tin protoporphyrin (SnPPIX). With the similar structure, the chemicals occupy the heme binding pocket of HO and competitively inhibit the enzyme activity [93]. Metalloporphyrin are widely used in HO-1-related studies and they have been confirmed with the efficiency against cancers. ZnPPIX was showed to inhibit the proliferation of leukemic cells [95] and suppress the resistance against panobinostat [96]. Treatment with pegylated-ZnPPIX significantly attenuated tumor growth and increased the sensitivity to chemotherapy in human colon cancer SW480 xenograft mouse model [97]. Besides, ZnPPIX significantly increased the chemotherapeutic response of cisplatin in hepatoma cells [98] and laryngeal squamous cancer cells [99], as well as the efficiency of photodynamic therapy in cultured melanoma tumor cells [100]. In vivo studies, ZnPPIX reduced tumor growth of LL/2 lung cancer in C57BL mice, by suppressing vascular endothelial growth factor concentration in tumors [101]. In addition, ZnPPIX inhibited peritoneal metastasis of gastric cancer via anti-angiogenesis and improved the survival of tumor-bearing mice, which were related to the suppression of ROS production and ERK activation [102]. However, the non-selective activity of metalloporphyrin limits its application, due to the similarity of structure to heme. Metalloporphyrins can affect other heme-containing enzymes, such as nitric oxide synthases (NOS), soluble guanylyl cyclase (sGCs), and cytochromes P450 (CYP450) [103,104]. Azole-based compounds are structurally distinctive from metalloporphyrins and highly selective for the HO-1 inhibition [91,94], exhibiting low or no inhibitory activity on NOS, sGCs, or CYP450 [104,105]. They bind to the distal side of heme through an azole anchor which coordinates with the heme iron [106]. Imidazole-derived HO-1 inhibitors showed a better profile in HO-1 inhibition in prostate and breast cancer cell lines [107]. A series of hybrid compounds of imatinib and imidazole-based HO-1 inhibitors were able to inhibit both tyrosine kinase and HO-1/HO-2 activities and reduced the viability in imatinib-resistant chronic myeloid leukemia cells [108].

Specific inhibition of HO-1 can be achieved through genetic tools. Small interfering RNA (siRNA) and short hairpin RNA (shRNA) are two common approaches on gene silencing via the cleavage of target gene or inhibition of protein synthesis. An intervention of HO-1 expression by shRNA was shown to induce apoptosis in lung cancer cells [109], colon carcinoma cells [45], leukemic cells [95,110], esophageal squamous carcinoma cells [111], and breast cancer cells [112]. In addition, siRNA/shRNA of HO-1 also enhanced the sensitivity to chemotherapy in pancreatic cancers [113,114], lung cancer [109], and breast cancer [112]. In an orthotopic model of hepatocellular carcinoma, intraperitoneal injection of HO-1 siRNA attenuated the growth of tumor [115]. Moreover, in the 5-fluorouracil resistant human colon cancer cells xenografted subcutaneously, HO-1 knockdown by shRNA significantly reduced tumor size and increased the sensitivity to 5-fluorouracil treatment [116]. Furthermore, the CRISPR/Cas9 knockout system also gave positive results, showing HO-1 depletion to mediate a decrease of 293T cell viability, growth, and an increase of sensitivity to H_2_O_2_ treatment [117].

## 9. Conclusion and Prospective

With the genetically heterogeneous and labyrinthine gene expression, anti-drug regulation of survival and metastatic metabolism of tumors against cancer treatment, most aggressive cancer treatments have less favorable outcomes, reflecting the lack of sufficient promising therapies that are capable of curing the most aggressive cancers. Chemotherapy is being developed against cancer with new chemotherapeutic drugs and strategies being tested in preclinical and clinical trials. In this review, we discussed how HO-1 regulates ferroptosis, the therapeutic strategy by manipulating HO-1 to mediate ferroptosis, and prospective chemotherapeutic drugs against cancer via HO-1-mediated ferroptosis. Several prospective chemodrugs (e.g., BAY117085, withaferin A, erastin, RSL3, and sorafenib) involved in the HO-1-mediated ferroptosis for chemotherapeutic strategy are discussed, providing some representative examples of the application in killing different types of cancer cells. Up to date, few studies have focused on the role of HO-1 in ferroptosis. Results from these studies implicate that the dual role of HO-1 in ferroptosis regulation might depend on different pathological conditions. The precise mechanism of this phenomenon needs to be further investigated. Although iron-dependent and ROS-promoted ferroptosis was redefined at 2012 [6], a large part of the mechanisms underlying the regulation by HO-1 remains elusive, particularly in respect to applications of chemotherapy in cancers.

## Figures and Tables

**Figure 1 ijms-20-00039-f001:**
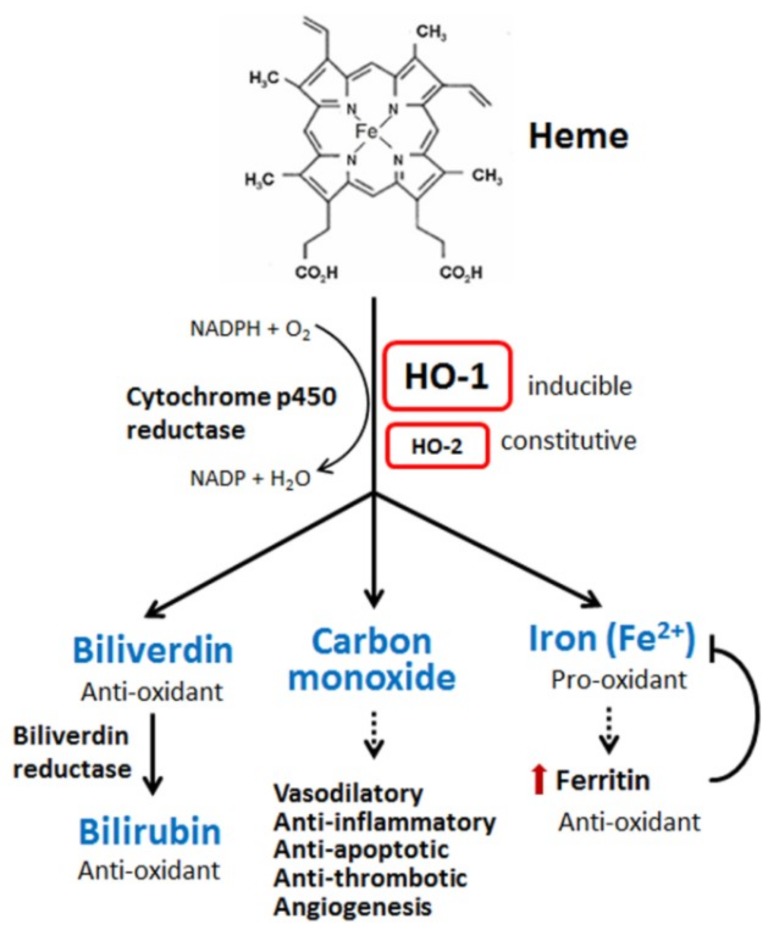
Heme metabolism. Heme is degraded by heme oxygenase (HO), leading to the generation of biliverdin, carbon monoxide, and ferrous iron. Biliverdin is subsequently converted to bilirubin by biliverdin reductase. Under most conditions, biliverdin and bilirubin act as anti-oxidants by scavenging or neutralizing reactive oxygen species (ROS). Carbon monoxide, a gaseous product, mainly functions in signaling transduction, including the vasodilation of blood vessels, production of anti-inflammatory cytokines, upregulation of anti-apoptotic effectors, and thrombosis. Ferrous iron is the major pro-oxidant in all metabolites of heme. However, heme oxygenase-1 (HO-1) activation also increases ferritin expression, which can bind to ferrous iron and detoxify its pro-oxidant effect. The black arrows indicate that biliverdin metabolize into bilirubin. The dotted arrow indicates that carbon monoxide serves a regulator in vasodilatory, anti-inflammatory, anti-apoptotic, anti-thromobtic, and angiogenesis activities. The dotted arrow below iron indicates the iron increase will increase ferritin, which neutralizes the pro-oxidant effect of iron.

**Figure 2 ijms-20-00039-f002:**
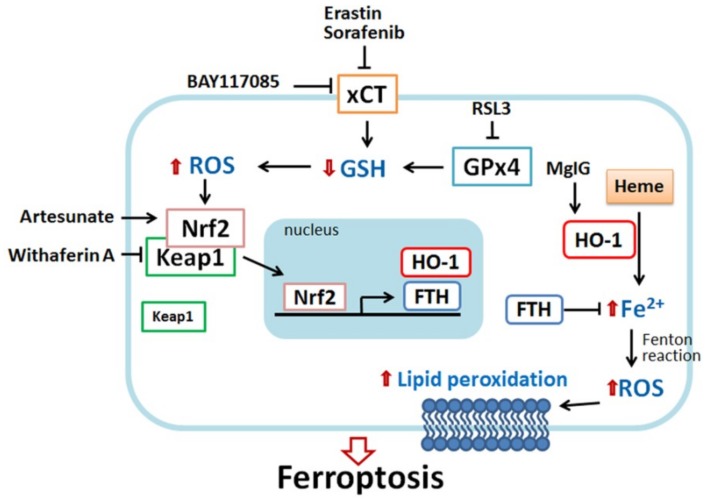
Scheme of HO-1-regulated ferroptosis. HO-1 plays a dual role in ferroptosis, pro-ferroptotic and anti-ferroptotic effects. Erastin and sorafenib (xCT inhibitor) and RSL3 (glutathione peroxidase 4 (GPx4) inhibitor) can deplete glutathione, leading to ROS generation. In response to oxidative stress, nuclear factor erythroid 2-related factor 2 (Nrf2) disassociates from Kelch-like ECH-associated protein 1 (Keap1), and then migrates into nuclei, where it binds the antioxidant-response element (ARE) site of target genes such as HO-1 and ferritin. HO-1 catalyzes heme degradation to generate ferrous iron (Fe^2+^). Ferrous iron is highly reactive as a pro-oxidant and, thus, produces ROS. Excessive ROS damage intracellular structures and DNA, causing the peroxidation of lipid and protein and eventually cell death. Nrf2 induces ferritin expression to chelate ferrous iron, avoiding ROS overload. Recently, some small molecules were identified to possess a pro-ferroptosis effect through HO-1. Heme can directly activate HO-1 expression. Similar to erastin and sorafenib, BAY117089 can deplete GSH and increase ROS production, resulting in Nrf2−HO-1 activation and ferroptosis. Withaferin A directly targets Keap1 and releases Nrf2, followed by HO-1 activation, iron accumulation, and cell death. Magnesium isoglycyrrhizinate (MgIG) increases HO-1 expression and free cellular iron level. By contrast, the activation of HO-1 might provide a cytoprotective effect. For example, in erastin-, sorafenib-, and RSL-stimulated cells, ferritin expression is increased through the Nrf2−HO-1 pathway and neutralize iron toxicity. Nrf2-targeted antioxidant gene expression also benefits the acquisition of drug resistance. Artesunate also induces the Nrf2−HO-1 signal to assist cells to acquire drug resistance.

**Figure 3 ijms-20-00039-f003:**
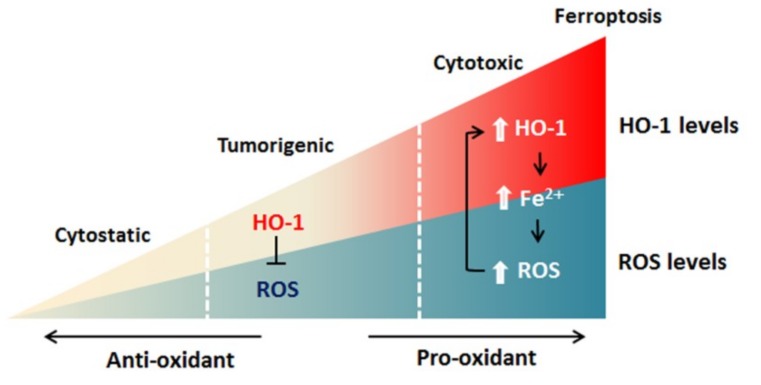
Model of HO-1-mediated ferroptosis. HO-1 exerts a cytoprotective effect by scavenging ROS during moderate activation. By contrast, excessive activation of HO-1 increases labile Fe^2+^, leading to ROS overload and death of cancer cells.

**Table 1 ijms-20-00039-t001:** Role of increased HO-1 in the ferroptosis and anti-ferroptosis response to different inducers.

Drugs	Cell Types	HO-1 Activity	Significance of HO-1 Increase in Ferroptosis	References
Heme	Platelets	Upregulated	Pro-ferroptotic effect ↑Iron ↑ROS ↓GSH ↑Lipid peroxidation	[66]
Hydrogen peroxide	A549 lung adenocarcinoma	Upregulated	Anti-ferroptotic effect ↑Transferrin ↑Transferrin receptor ↑Ferritin ↓Iron	[57]
Erastin RSL3	HT-1080 fibrosarcoma cells	Upregulated	Pro-ferroptotic effect ↑Iron ↑CO	[15]
Erastin Sorafenib	Hepatocellular carcinoma, HepG2 and Hepa1-6 cells	Upregulated	Anti-ferroptotic effect ↑ROS ↑Iron ↑Ferritin	[31]
Magnesium isoglycyrrhizinate	CCL4-induced liver fibrosis rat model Hepatic stellate cell line HSC-T6	Upregulated	Pro-ferroptotic effect ↑Iron ↑Lipid peroxidation	[29]
Artesunate	Cisplatin-resistant head and neck cancer cell	Upregulated	Anti-ferroptotic effect HO-1 co-works with NQO-1 to serve as antioxidant	[22]
BAY117085	Triple-negative breast cancer, MDA-MB-231 cells; *Glioblastoma, DBTRG-05MG*	Upregulated	Pro-ferroptotic effect Inhibit system Xc^−^ ↓GSH ↑ROS ↑Iron	[16]
Withaferin A	Neuroblastoma, IMR-32 cells.	Upregulated	Pro-ferroptotic effect ↑ROS ↑Iron	[17]

The up arrows indicate the increased expression or enhanced activity. The down arrows indicate decreased expression or decreased levels.

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
