# Peer review of "A Dual Role of Heme Oxygenase-1 in Cancer Cells"

_ijms, 2018, doi:10.3390/ijms20010039_

Round 1

Reviewer 1 Report

The review is of high interest to the readers and the authors demonstrate knowledge on the subject. The review is extensive but well organized. My major concern is the English used throughout the text. The use of very large phrases, wrong grammar, and the abuse of definite/indefinite articles make the text very hard to read. This limits greatly the ability of the readers to understand this subject.

Scientifically, the review is solid. There are some minor errors:

- Page 4, Line 41: "Iron joins in the transfer of electrons..." should read "Iron is involved in the transfer of electrons..."

- Page 4, Line 44: "Iron binds to transferrin in a reduced ferric state (Fe3+)." should read "Iron binds to transferrin in an oxidized ferric state (Fe3+)."

- Page 7, Line 22:"However, because of the ability to shuttle electrons between protein and iron, 

heme-containing enzymes can catalyze both reductive and oxidative reactions." should read "However, because of the ability of iron to have different oxidation statesheme-containing enzymes can catalyze both reductive and oxidative reactions." 

- Page 9, Line 4: "...isolated from Artemisia annual L.,..." should read "...isolated from Artemisia annua L.,..."

Author Response

Reviewer 1

Comments and Suggestions for Authors

The review is of high interest to the readers and the authors demonstrate knowledge on the subject. The review is extensive but well organized. My major concern is the English used throughout the text. The use of very large phrases, wrong grammar, and the abuse of definite/indefinite articles make the text very hard to read. This limits greatly the ability of the readers to understand this subject.

[Our response] Thanks for your kind suggestion. At your request, we had our manuscript revised and grammar checked by Elsevier WebShop.

Scientifically, the review is solid. There are some minor errors:

1.      - Page 4, Line 41: "Iron joins in the transfer of electrons..." should read "Iron is involved in the transfer of electrons..."

[Our response] We corrected it.

2.      - Page 4, Line 44: "Iron binds to transferrin in a reduced ferric state (Fe3+)." should read "Iron binds to transferrin in an oxidized ferric state (Fe3+)."

[Our response] We corrected it.

3.      - Page 7, Line 22:"However, because of the ability to shuttle electrons between protein and iron, heme-containing enzymes can catalyze both reductive and oxidative reactions." should read "However, because of the ability of iron to have different oxidation states, heme-containing enzymes can catalyze both reductive and oxidative reactions." 

[Our response] We corrected it.

4.      - Page 9, Line 4: "...isolated from Artemisia annual L.,..." should read "...isolated from Artemisia annua L.,..."

[Our response] We corrected it.

Reviewer 2 Report

The present review deal with the “dark side of heme oxygenase-1 and its implication for cancer chemotherapy”. The subject is interesting and worth of investigating; however, in my opinion, there are not enough experimental data to correlate HO-1 overexpression to ferroptosis and to cancer therapy. The review is well written but is poorly informative and the connection between cancer, HO-1 and its metabolites, and ferroptosis is poorly supported and explained. The authors should discuss more in details the literature relevant to the involvement of HO-1 overexpression, ferroptosis and cancer. The authors should explain the interconnection more in details and demonstrate their theory more pragmatically since a major body of evidence suggest protective roles for HO-1 in cancer therefore the need of inhibiting HO-1.

In addition, it is strongly suggested that the authors should dedicate a paragraph describing the large body of evidence in which HO-1 inhibition (by knockout animals and azole-based inhibitors) is used as a valuable approach in the management of cancer therapy.

The authors discuss the “cytoprotective or detrimental effects of heme metabolites”, namely biliverdin/bilirubin and CO but not in relation with ferroptosis and cancer. The discussion should be relevant to the subject.

The authors should revise and improve the English form.

Finally, references should be updated and added where there is a lack. E.g. Pag 2 lines 1-4 please add references; pag. 2 lines 10-11 please add references; and so on and so forth.

Author Response

Reviewer 2

Comments and Suggestions for Authors

The present review deal with the “dark side of heme oxygenase-1 and its implication for cancer chemotherapy”. The subject is interesting and worth of investigating; however, in my opinion, there are not enough experimental data to correlate HO-1 overexpression to ferroptosis and to cancer therapy. The review is well written but is poorly informative and the connection between cancer, HO-1 and its metabolites, and ferroptosis is poorly supported and explained.

1.      The authors should discuss more in details the literature relevant to the involvement of HO-1 overexpression, ferroptosis and cancer. The authors should explain the interconnection more in details and demonstrate their theory more pragmatically since a major body of evidence suggest protective roles for HO-1 in cancer therefore the need of inhibiting HO-1.

[Our response] Thanks for your kind suggestion. The connection between HO-1 and cancer has been thoroughly discussed in many reviewed paper. The aim of this paper is to give an introduction regarding HO-1 in ferroptosis regulation correlated to cancer. The iron-dependent cell death, ferroptosis, was just re-defined in 2012. Unfortunately, so far there were only a few papers focused on the cancer regulation by the HO-1-associated ferroptosis. However, we were still trying to write some new paragraphs to discuss the connection between cancer, HO-1, and its metabolites, and ferroptosis at your request, which was shown in red. We believed that more research on ferroptosis will give more insights into HO-1-regulated ferroptosis in cancer progression and metabolism in the near future.     

2.      In addition, it is strongly suggested that the authors should dedicate a paragraph describing the large body of evidence in which HO-1 inhibition (by knockout animals and azole-based inhibitors) is used as a valuable approach in the management of cancer therapy.

[Our response] At your request, we added the supporting evidence of HO-1 connected to ferroptosis from experimental approaches of HO-1, such as knockdown and inhibitors of HO-1.

3.      The authors discuss the “cytoprotective or detrimental effects of heme metabolites”, namely biliverdin/bilirubin and CO but not in relation to ferroptosis and cancer. The discussion should be relevant to the subject.

[Our response] The biliverdin/bilirubin was less connected with ferroptosis. So far, there was only one paper on it. We added the results of the paper into this review.

4.      The authors should revise and improve the English form.

[Our response] At your request, we had our manuscript revised and grammar checked by Elsevier WebShop.

5.      Finally, references should be updated and added where there is a lack. E.g. Pag 2 lines 1-4 please add references; pag. 2 lines 10-11 please add references; and so on and so forth.

[Our response] We added the references to our manuscript. 

Reviewer 3 Report

In their manuscript the authors discuss the actual literature about the prooxidant role of HO-1. The topic is up-to-date and interesting. Unfortunately, the possible applications of HO-1 mediated ferroptosis in the treatment of cancer relies only on different cancer cell lines. The key question could be that how to reach the induction of HO-1 mediated ferroptosis only in cancer cells and not in the intact cells. Maybe the author should discuss this important problem in a short paragraph. Furthermore, what is the crucial factor, which determines that Nrf2-HO-1 pathway will function as antioxidant or prooxidant?

Finally, this reviewer suggests to ask a native English speaker or an official English editing service to check the grammar.

Author Response

Reviewer 3

Comments and Suggestions for Authors

In their manuscript the authors discuss the actual literature about the prooxidant role of HO-1. The topic is up-to-date and interesting. Unfortunately, the possible applications of HO-1 mediated ferroptosis in the treatment of cancer relies only on different cancer cell lines. The key question could be that how to reach the induction of HO-1 mediated ferroptosis only in cancer cells and not in the intact cells. Maybe the author should discuss this important problem in a short paragraph. Furthermore, what is the crucial factor, which determines that Nrf2-HO-1 pathway will function as antioxidant or prooxidant?

[Our response] Thanks for your kind suggestion. We created several paragraphs to discuss the antioxidant or pro-oxidant role of HO-1 in cancer cells and normal cells. These new paragraphs were marked in red.

Finally, this reviewer suggests to ask a native English speaker or an official English editing service to check the grammar.

[Our response] At your request, we had our manuscript revised and grammar checked by Elsevier WebShop.

Reviewer 4 Report

The authors present a well-written and clear review about the multi-faced roles of HMOX1 in health and disease. I recommend the article for publication and only have minor suggestions:

Include the option of NRF2-independent HMOX1 activation through de-activation of BACH1. You have mentioned this in line 22 in a slightly different context but did not dwell on the implications of the study of Reichard et al. Nucleic Acid Research (2007).

In the context of cancer, it may be useful to refer to a recent study  showing its upregulation in 8 human cancer cell lines in response to oxidative stress when talking about ROS+cancer+HO-1 in some parts of your manuscript, see DOI 10.3390/antiox7110151

Minor spell check / linguistic style check required, e.g. in Table 1 maybe use "upregulated" instead of "upregulate"

Author Response

Reviewer 4

Comments and Suggestions for Authors

The authors present a well-written and clear review about the multi-faced roles of HMOX1 in health and disease. I recommend the article for publication and only have minor suggestions:

Include the option of NRF2-independent HMOX1 activation through de-activation of BACH1. You have mentioned this in line 22 in a slightly different context but did not dwell on the implications of the study of Reichard et al. Nucleic Acid Research (2007).

[Our response] Thanks for your suggestion. We replaced the reference.

In the context of cancer, it may be useful to refer to a recent study  showing its upregulation in 8 human cancer cell lines in response to oxidative stress when talking about ROS+cancer+HO-1 in some parts of your manuscript, see DOI 10.3390/antiox7110151

[Our response] Thanks for your suggestion. We added this study to our manuscript.

Minor spell check / linguistic style check required, e.g. in Table 1 maybe use "upregulated" instead of "upregulate"

[Our response] We corrected it. At your request, we had our manuscript revised and grammar checked by Elsevier WebShop.

Round 2

Reviewer 2 Report

Comments and Suggestions for Authors

Second Revision: My response to the authors are embedded along the text.

The present review deal with the “dark side of heme oxygenase-1 and its implication for cancer chemotherapy”. The subject is interesting and worth of investigating; however, in my opinion, there are not enough experimental data to correlate HO-1 overexpression to ferroptosis and to cancer therapy. The review is well written but is poorly informative and the connection between cancer, HO-1 and its metabolites, and ferroptosis is poorly supported and explained.

1.      The authors should discuss more in details the literature relevant to the involvement of HO-1 overexpression, ferroptosis and cancer. The authors should explain the interconnection more in details and demonstrate their theory more pragmatically since a major body of evidence suggest protective roles for HO-1 in cancer therefore the need of inhibiting HO-1.

[Our response] Thanks for your kind suggestion. The connection between HO-1 and cancer has been thoroughly discussed in many reviewed paper. The aim of this paper is to give an introduction regarding HO-1 in ferroptosis regulation correlated to cancer. The iron-dependent cell death, ferroptosis, was just re-defined in 2012. Unfortunately, so far there were only a few papers focused on the cancer regulation by the HO-1-associated ferroptosis. However, we were still trying to write some new paragraphs to discuss the connection between cancer, HO-1, and its metabolites, and ferroptosis at your request, which was shown in red. We believed that more research on ferroptosis will give more insights into HO-1-regulated ferroptosis in cancer progression and metabolism in the near future.     

Response of the reviewer:

Since the role of “HO-1 in ferroptosis regulation correlated to cancer” as the authors itself agreed is mentioned in a ridiculous number of papers the review in the present form, even though very well written, is just speculative. The readers that are not familiar with the subject can be misled and the title itself of the review is misleading. Therefore, I suggest changing the title since there is no consolidated evidence of a direct correlation between induction/expression HO-1, ferroptosis and cancer. I strongly suggest to avoid any mention to cancer in the title.

2.      In addition, it is strongly suggested that the authors should dedicate a paragraph describing the large body of evidence in which HO-1 inhibition (by knockout animals and azole-based inhibitors) is used as a valuable approach in the management of cancer therapy.

[Our response] At your request, we added the supporting evidence of HO-1 connected to ferroptosis from experimental approaches of HO-1, such as knockdown and inhibitors of HO-1.

 Response of the reviewer:

The authors added section 3. “HO-1 and ferroptosis” but it was not suggested to add a section putting in relation HO-1 and ferroptosis but a section putting in relation HO-1 inhibition and the management of cancer therapy.

In addition, the authors added section “7. Manipulation of HO-1 in ferroptosis for cancer treatment” That is, once more, a pure speculative exercise since they did not discuss any direct correlation between HO-1 induction, ferroptosis and cancer. In fact, the authors cite only general review dealing with ROS levels and one very old publication (1999).

Therefore, once more the author are strongly encouraged to give the reader a more exhaustive idea of the role of HO-1 in cancer by means of knockout animals and azole-based inhibitors. The authors should add a new section dedicated to the discussion of the work of the research group of Kanji Nakatsu that developed azole-based HO-1 inhibitors and the work of prof. Salerno continuing very rencently on the development of HO-1 inhibitors and their application in cancer therapy. This large body of information cannot be ignored to give the readers an exhaustive view of the contradicting role of HO-1 in cancer.

Please find hereafter some relevant papers:

Robert Kinobe, Ryan A Dercho, Kanji Nakatsu. Inhibitors of the heme oxygenase - Carbon monoxide system: On the doorstep of the clinic? Canadian Journal of Physiology and Pharmacology 86(9):577-99  

Salerno L., Amata E., Romeo G., Marrazzo A., Prezzavento O., Floresta G., Sorrenti V., Barbagallo I., Rescifina A., Pittala, V. Potholing of the hydrophobic heme oxygenase-1 western region for the search of potent and selective imidazole-based inhibitors. Eur. J. Med. Chem. 2018, 148, 54-62

Salerno L., Romeo G., Modica M.N., Amata E., Sorrenti E., Barbagallo I., Pittalà V. Heme Oxygenase-1: a New Druggable Target in the Management of Chronic and Acute Myeloid Leukemia. Eur. J. Med. Chem. 2017, 142, 163-178

Loboda A1, Jozkowicz A2, Dulak J3. HO-1/CO system in tumor growth, angiogenesis and metabolism - Targeting HO-1 as an anti-tumor therapy. Vascul Pharmacol. 2015, 74:11-22.

Amata, E., Marrazzo, A., Dichiara, M., (...), Romeo, G., Pittalà, V. Heme Oxygenase Database (HemeOxDB) and QSAR Analysis of Isoform 1 Inhibitors. ChemMedChem 2017, 12(22), pp. 1873-1881

3.      The authors discuss the “cytoprotective or detrimental effects of heme metabolites”, namely biliverdin/bilirubin and CO but not in relation to ferroptosis and cancer. The discussion should be relevant to the subject.

[Our response] The biliverdin/bilirubin was less connected with ferroptosis. So far, there was only one paper on it. We added the results of the paper into this review.

4.      The authors should revise and improve the English form.

[Our response] At your request, we had our manuscript revised and grammar checked by Elsevier WebShop.

5.      Finally, references should be updated and added where there is a lack. E.g. Pag 2 lines 1-4 please add references; pag. 2 lines 10-11 please add references; and so on and so forth.

[Our response] We added the references to our manuscript. 

Author Response

Responses to Reviewer 2

1.     Since the role of “HO-1 in ferroptosis regulation correlated to cancer” as the authors itself agreed is mentioned in a ridiculous number of papers the review in the present form, even though very well written, is just speculative. The readers that are not familiar with the subject can be misled and the title itself of the review is misleading. Therefore, I suggest changing the title since there is no consolidated evidence of a direct correlation between induction/expression HO-1, ferroptosis and cancer. I strongly suggest to avoid any mention to cancer in the title.

[Our response] Thank you for the suggestion. The title was replaced with: A dual role of heme oxygenase-1 in cancer cells”.

 2.     The authors added section 3. “HO-1 and ferroptosis” but it was not suggested to add a section putting in relation HO-1 and ferroptosis but a section putting in relation HO-1 inhibition and the management of cancer therapy.

In addition, the authors added section “7. Manipulation of HO-1 in ferroptosis for cancer treatment” That is, once more, a pure speculative exercise since they did not discuss any direct correlation between HO-1 induction, ferroptosis and cancer. In fact, the authors cite only general review dealing with ROS levels and one very old publication (1999).

Therefore, once more the author are strongly encouraged to give the reader a more exhaustive idea of the role of HO-1 in cancer by means of knockout animals and azole-based inhibitors. The authors should add a new section dedicated to the discussion of the work of the research group of Kanji Nakatsu that developed azole-based HO-1 inhibitors and the work of prof. Salerno continuing very rencently on the development of HO-1 inhibitors and their application in cancer therapy. This large body of information cannot be ignored to give the readers an exhaustive view of the contradicting role of HO-1 in cancer.

Please find hereafter some relevant papers:

Robert Kinobe, Ryan A Dercho, Kanji Nakatsu. Inhibitors of the heme oxygenase - Carbon monoxide system: On the doorstep of the clinic? Canadian Journal of Physiology and Pharmacology 86(9):577-99.

Salerno L., Amata E., Romeo G., Marrazzo A., Prezzavento O., Floresta G., Sorrenti V., Barbagallo I., Rescifina A., Pittala, V. Potholing of the hydrophobic heme oxygenase-1 western region for the search of potent and selective imidazole-based inhibitors. Eur. J. Med. Chem. 2018, 148, 54-62.

Salerno L., Romeo G., Modica M.N., Amata E., Sorrenti E., Barbagallo I., Pittalà V. Heme Oxygenase-1: a New Druggable Target in the Management of Chronic and Acute Myeloid Leukemia. Eur. J. Med. Chem. 2017, 142, 163-178.

Loboda A1, Jozkowicz A2, Dulak J3. HO-1/CO system in tumor growth, angiogenesis and metabolism - Targeting HO-1 as an anti-tumor therapy. Vascul Pharmacol. 2015, 74:11-22.

Amata, E., Marrazzo, A., Dichiara, M., (...), Romeo, G., Pittalà, V. Heme Oxygenase Database (HemeOxDB) and QSAR Analysis of Isoform 1 Inhibitors. ChemMedChem 2017, 12(22), pp. 1873-1882.

[Our response] Section 3 was replaced with “HO-1 and cancel cell survival: and Section 7 with Manipulation of HO-1 in ferroptosis”. We also added one paragraph, section 8 “HO-1 modulators in cancer treatment”. 

Round 3

Reviewer 2 Report

The authors addressed the reviewer comment. One only minor amendations is required:

Abstract:

Heme oxygenase (HO) -1 is known to metabolize hemes into biliverdin/bilirubin, carbon monoxide, and ferrous iron and has been suggested for cytoprotective effects in various stress-related conditions. However, increasing studies showed a dark side of HO-1, in which HO-1 acts as a critical mediator in ferroptosis induction and plays a causative factor for the pathological progress of many diseases.

SHOULD BE CHANGED INTO:

Heme oxygenase (HO) -1 is known to metabolize hemes into biliverdin/bilirubin, carbon monoxide, and ferrous iron and has been suggested for cytoprotective effects in various stress-related conditions. HO-1 is commonly regarded as a survival molecule, exerting an important role in cancer progression and its inhibition is considered beneficial in a number of cancers. However, increasing studies showed a dark side of HO-1, in which HO-1 acts as a critical mediator in ferroptosis induction and plays a causative factor for the pathological progress of many diseases.

Author Response

[Our response] Thank you for your suggestion. We changed the abstract as your suggestion. The change was marked in red.